# Does the Locally-Adaptive Model of Archaeological Potential (LAMAP) work for hunter-gatherer sites? A test using data from the Tanana Valley, Alaska

Rob Rondeau[1]☯, W. Christopher Carleton[2]☯, Mark Collard[1]‡, Jonathan Driver[1]‡*

**1** Department of Archaeology, Simon Fraser University, Burnaby, British Columbia, Canada, **2** Max Planck Institute for Chemical Ecology, Jena, Germany

☯ These authors contributed equally to this work.
‡ These authors also contributed equally to this work.
* driver@sfu.ca

**Data Availability Statement:** All relevant data are within the manuscript and its Supporting information files, or via links within those files. Note that precise data on the location of

## Abstract

We report an assessment of the ability of the Locally-Adaptive Model of Archaeological Potential (LAMAP) to estimate archaeological potential in relation to hunter-gatherer sites. The sample comprised 182 known sites in the Tanana Valley, Alaska, which was occupied solely by hunter-gatherers for about 14,500 years. To estimate archaeological potential, we employed physiographic variables such as elevation and slope, rather than variables that are known to vary on short time scales, like vegetation cover. Two tests of LAMAP were carried out. In the first, we used the location of a random selection of 90 sites from all time periods to create a LAMAP model. We then evaluated the model with the remaining 92 sites. In the second test, we built a LAMAP model from 12 sites that pre-date 10,000 cal BP. This model was then tested with sites that post-date 10,000 cal BP. In both analyses, areas predicted to have higher archaeological potential contained higher frequencies of validation sites. The performance of LAMAP in the two tests was comparable to its performance in previous tests using archaeological sites occupied by agricultural societies. Thus, the study extends the use of LAMAP to the task of estimating archaeological potential of landscapes in relation to hunter-gatherer sites.

## Introduction

Identifying areas of a landscape that have a high potential of containing archaeological sites is an important task for archaeologists. There are two main reasons for this. One is that we need either complete documentation or representative samples of surviving sites in a region in order to understand settlement patterns and processes, and to document the variety of human behaviours in particular times and places. Additional data on the age, location, size, and function of sites provide a more comprehensive picture of the distribution of human activities across the landscape. Once regional surveys are complete, we can investigate site distribution

archaeological sites is not being released, in order to protect sites from vandalism. Those data are accessible to qualified researchers from Alaska Heritage Resources Survey, the State of Alaska's Office of History and Archaeology http://dnr.alaska. gov/parks/oha/ahrs/ahrs.htm.

**Funding:** MC: Canada Research Chairs Program (228117 and 231256) https://www.chairs-chaires. gc.ca/home-accueil-eng.aspx MC: Canada Foundation for Innovation (203808) https://www. innovation.ca/ MC: British Columbia Knowledge Development Fund (862-804231) https://www2. gov.bc.ca/gov/content/governments/technology-innovation/bckdf MC: Social Sciences and Humanities Research Council of Canada (SSHRC) (895-2011-1009) https://www.sshrc-crsh.gc.ca/ home-accueil-eng.aspx RR: Social Sciences and Humanities Research Council of Canada (SSHRC) (004-74800-19710) https://www.sshrc-crsh.gc.ca/ home-accueil-eng.aspx RR: Simon Fraser University Graduate Studies (14518) https://www. sfu.ca/gradstudies.html The funders had no role in study design, data collection and analysis, decision to publish, or preparation of the manuscript.

**Competing interests:** The authors have declared that no competing interests exist.

[1], demography [2], spatial networks [3], economic organisation [4], trade and transportation networks [5], core and periphery relationships [6], viewscapes [7], and other phenomena that have a spatial dimension.

The other reason for identifying areas of high archaeological potential relates to the protection and management of cultural heritage. Needless to say, preventing or mitigating damage and destruction of archaeological sites requires knowledge of where sites are likely to be located. Ideally, cultural heritage managers will allocate limited resources efficiently, by developing models that predict areas of high potential for site discovery. This is especially important in regions where archaeological sites are difficult to find using remote sensing, and where time-consuming and expensive foot-survey and sub-surface testing are usually required to discover archaeological remains. In addition, accurate predictive models can assist modern land developers in avoiding areas of high archaeological potential, thus reducing the time and money costs of archaeological mitigation [8–11].

Predicting archaeological potential has been an active area of methodological research for a long time [11, 12] and a range of statistical methods have become established in the literature [13–15]. Most methods involve regression models designed to estimate the probability of finding archaeological material given known site locations and a set of predictor variables. These approaches include logistic models, Generalised Additive Models (GAM), and Bayesian techniques [11].

Regression-based approaches suffer from some major drawbacks, however. One involves the need for non-site locations in order to fit the statistical models to the available data. Unfortunately, we almost never have reliable information about where sites are not located, which has led most scholars to use pseudo-absence data, i.e., randomly selected locations intended to represent non-site locations. A substantial problem with this pseudo-absence approach is that a randomly selected location may in fact have archaeological material and could have been a suitable location for use in the past. Consequently, any regression model applied to such data is very likely to have trouble distinguishing high and low potential locations. Recently, the impact of this flaw has been demonstrated quantitatively [15].

Another major drawback with the regression-based approaches is that they treat sites like points on the landscape instead of areas with internal variability. This is necessary for standard regression models where the dependent variable (e.g., site presence, count, or density) needs to be compared in a point-wise manner to independent variables (e.g., environmental covariates like elevation, slope, etc.). That is, one observation about the archaeological record must be compared to one observation about the landscape. Archaeological sites, however, occupy areas. Importantly, the landscape variability within an area may have been relevant for land-use decisions and, therefore, could be useful for discriminating high potential locations from low potential ones.

With these problems in mind, Carleton et al. [16] developed an alternate approach to predicting archaeological potential called the Locally-Adaptive Model of Archaeological Potential or LAMAP. An algorithm rather than single model, LAMAP differs from existing methods in that it does not depend on having 'non-site locations' to make predictions and does not treat archaeological sites like points on a map. Instead, the algorithm considers distributions of values for landscape variables around known sites (i.e., the training data) and then uses these distributions to estimate the archaeological potential of a given target location. LAMAP's results highlight landscape features in the study region that were important to people in the past, and this gives us a better idea about where to look for other areas that may have been utilised.

LAMAP shares features with another better known approach, MaxEnt. MaxEnt was developed in the field of ecology for species distribution modelling from presence-only data [17]. It, too, is an algorithm of sorts and involves comparing distributions of landscape variables rather

than estimating parameters for statistical models. However, MaxEnt still treats individual sites like points and compares a pooled distribution of those point data to the 'background' distribution of landscape variables estimated by randomly sampling point locations within a given study area. LAMAP differs from MaxEnt in that it involves comparing each location in a target prediction region with the distributions of landscape variables for every known site individually. These individual similarity measures are then combined to produce a final potential estimate for a given target location.

Two tests of the reliability of LAMAP have been published, to date. Carleton et al. [18] tested LAMAP with data from the Classic Maya region. They used the locations of 69 known Classic Period sites to produce a high-resolution predictive model for a 280 km$^2$ study area around the large Classic Maya centre of Minanha in west-central Belize. Next, pedestrian surveys were conducted in sample areas that represented the various LAMAP classes. Although new sites were found, this first test produced too little data to establish statistical significance, due to the problems of surveying in densely vegetated areas. Subsequently, Carleton et al. [18] utilised LIDAR imaging to identify features and structures, and to classify them into various settlement types. The results of this analysis revealed a strong relationship between LAMAP classes and the number of sites discovered through LIDAR image analysis. There were more than three times more sites in areas that were deemed to be high potential by LAMAP than in areas that the model classified as low potential.

The second test of LAMAP was reported by Willett et al. [19] and focused on the intensively studied Sagalassos region of southwestern Turkey. The researchers built seven LAMAP predictive models, using data for sites from 6000 BC to the modern era, and then combined the scores for each cell to create an aggregate archaeological potential score. Next, pedestrian surveys were conducted in previously unsurveyed locations that were selected to cover the full range of archaeological potential classes, with both structures and artifact scatters being treated as sites. LAMAP's success in identifying high potential areas varied from one time-period to another, but for every time-period its predictions of high potential were validated by survey results. Both the count of sites and the count of artifact scatters were higher as the LAMAP class increased.

While LAMAP has shown promise as a method for identifying high potential areas for finding archaeological sites, its use has so far been limited to sedentary agricultural societies, which means its general utility has yet to be established. Most notably, the method has not yet been applied to the problem of assessing archaeological potential in relation to sites produced by hunter-gatherers. We cannot assume LAMAP will work as well for hunter-gatherer sites as it does for the sites produced by agriculturalists because many prehistoric hunter-gatherer groups were highly mobile and often only used sites briefly. As such, their decisions about where to locate a site may have depended more on immediate circumstances and less on long-term considerations. These sorts of land-use differences between hunter-gatherers and sedentary agriculturalists could, in theory, pose a problem for LAMAP.

If LAMAP can be shown to identify areas that have a high potential for locating hunter-gatherer sites, the benefits will be considerable. Our ancestors and collateral relatives were dependent on wild resources and highly mobile for most of the last seven million years. This means that much of the archaeological record and some of the most important research topics in archaeology (e.g., early human expansion out of Africa, the peopling of nearly every continent) require the study of hunter-gatherer sites. A new, reliable method for predicting archaeological potential in connection with hunter-gatherer sites should, therefore, improve understanding of our collective human past. In addition, there are still large areas of the earth's surface (e.g., the boreal forest regions of North America) that have only ever been occupied by hunter-gatherers and where archaeological sites are difficult to find. Predictive models are essential for heritage management in such environments [9, 20–22].

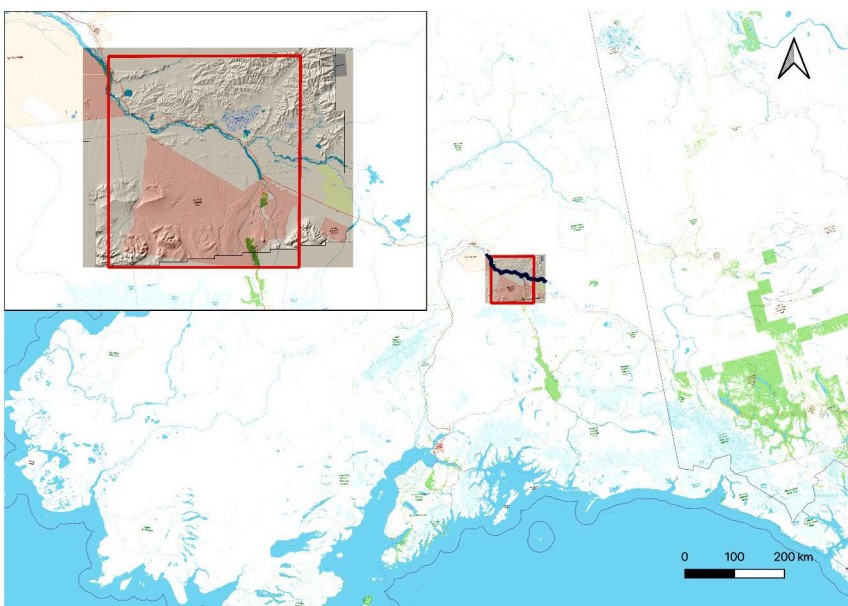

**Fig 1. Location of Tanana Valley study area in Alaska.** The data used to produce the background layers in this image are available under the Open Database License from OpenStreetMap.org (https://www.openstreetmap.org/copyright).

In this paper, we report the first attempt to evaluate LAMAP's ability to estimate archaeological potential in relation to hunter-gatherer sites. Our test region consists of 7,000 km$^2$ of the valley of the Tanana River in central Alaska (Fig 1). The study area contains a 14,500-year-long sequence of hunter-gatherer archaeological sites [23, 24], many of which can be dated through radiocarbon or typological cross-dating. The region has experienced minimal disturbance by modern industry and, unlike many parts of the world, there is no overlay of agricultural settlements to obscure earlier hunter-gatherer sites.

We addressed two questions in the study. First, can LAMAP be used to identify areas within the Tanana Valley that have a high potential of containing hunter-gatherer sites? To answer this question, we carried out an analysis using cross-validation. Half of the known sites were randomly selected to use as a training dataset and the remaining known sites were used to test the model. We reasoned that if the LAMAP method is suitable for identifying high potential areas for hunter-gatherer sites, there should be a positive correlation between LAMAP predictive classes and the number of cross-validation sites associated with those predictions.

Second, we asked whether a model built on the location of sites occupied during one period can predict areas of high potential for sites from a different time-period. We divided the sample of known sites into pre-10,000 cal BP sites and post-10,000 cal BP sites. We then used the pre-10,000 cal BP sites to train the model and the post-10,000 cal BP sites to test it. This analysis allowed us to investigate whether site location preferences shifted over time or whether the underlying reasons for hunter-gatherer site location decisions were so consistent that they transcended the environmental changes associated with the Pleistocene/Holocene transition.

## Background

### LAMAP

LAMAP has been described in detail in two earlier publications [16, 18]. Briefly, the model's key assumption is that known-site locations can provide us with insight regarding the

geographic and ecological characteristics of places used by people in the past. If people spent time at a location, as evidenced by material culture remains, this implies that the landscape around the site was suitable for whatever activity was taking place there. For example, if a known site appears to have been a base camp it is reasonable to assume that the location was suitable for base camp activities. This implies that other areas possessing the same physiographic and ecological characteristics were probably also suitable for use as a base camp. In other words, parts of the landscape that are similar to landscapes around known sites were probably selected on other occasions. Thus, without knowing exactly why humans made certain land-use decisions in the past or even what precisely people were doing in a given place, we can estimate the archaeological potential of any unsurveyed location, given known-site locations as a basis for comparison. This 'known-site suitability assumption' is the foundation of the LAMAP approach.

LAMAP assesses archaeological potential by comparing unsurveyed areas in a target region to circular sampling areas (CSAs) of 1 km diameter around known-sites. More specifically, it uses empirical data to find areas in the target region that are similar to the CSAs around known-sites with respect to a set of landscape variables. Ideally, the landscape variables represent persistent landscape characteristics that change relatively slowly so that measurements at different locations would have remained relatively constant with respect to each other through time. It is also preferable to use variables for which data are readily available, such as derivatives of physiography (e.g. elevation and slope), ecological and environmental observations (e.g. vegetation or soil type), or the product of computation (e.g. distances to water or intervisibility estimates).

LAMAP employs a relatively simple algorithm to estimate the archaeological potential of unsurveyed areas. The algorithm scans a raster image for the study area (i.e., a grid of cells that represents the study area) and assigns a probability value to every cell in the image. The probabilities are derived from the distributions of landscape variables around a set of known site locations. These distributions are based on the cells in the raster image located in the CSAs around the known-sites. Each cell contains a value for a given landscape variable, such as the elevation value of the terrain in meters above sea level at the corresponding location on the ground. Together, the cells then represent the distribution of values for the relevant landscape variable around the known-site locations. With these distributions, the algorithm can then estimate the empirical probability of finding a target cell with a given value for a given landscape variable that is similar to the corresponding values found in CSAs around all known-sites. The estimate is effectively the empirical likelihood of finding a place in a target region that is similar to the landscape around known archaeological sites.

LAMAP's estimates of archaeological potential are probabilities. They are calculated on a cell-by-cell basis to produce a LAMAP raster surface. A LAMAP value for a given target cell answers the question: what is the probability of finding a cell within the CSA around a known-site that is similar to the target cell? The LAMAP algorithm uses empirical frequency distributions of measurements for the relevant variables around a given known-site location to estimate the probability. Thus, for six landscape variables, there would be six probabilities for a single comparison between one target cell and one known-site location. These probabilities are multiplied to give the probability of finding a spot like the target cell in the area around the known-site. The same calculation is performed for every known-site, producing a list of probabilities for a given target cell. Each element of the list is the LAMAP estimate comparing the target cell and one known-site location. Every cell in the study area is compared to each known-site CSA. Then, the probabilities in the list are weighted according to the distance between the target cell and the relevant known-site. Lastly, the weighted probabilities are combined using the 'Law of Total Probability', a formula for determining the overall probability of

an outcome (in this case, find similar spots in all known-sites) that can be realised by multiple distinct events (in this case, find a similar spot at one site). The calculation results in a value ranging from 0 to 1 for every cell in the study area. The higher the value for a given cell, the more similar it is to cells around known sites, and therefore the higher its archaeological potential. The estimates can then be binned into categories (quantiles) and re-classified into an ordinal variable to facilitate evaluation of the model's utility. These ordinal values indicate relative potential, with higher values indicating higher archaeological potential. In theory, areas of cells within the unsurveyed areas with the highest LAMAP values should contain more sites than areas with lower LAMAP values.

Prior to computing the LAMAP values, the landscape variables are transformed with Principal Component Analysis (PCA) to facilitate faster computation and improve the contrast between locations. PCA is a dimension reduction technique that is usually used to emphasise variation and reduce redundancy among variables in an analysis [25]. It takes a set of potentially correlated variables, such as elevation, slope, and distance to drainages, and reduces them to a new set of uncorrelated ones. Fewer variables result in faster calculations. The PCA also helps to avoid rounding or floating-point computational limits. Computers have a limited number of decimal places to work with, so multiplying small probabilities by other small probabilities can lead to false zeros. Perhaps most importantly, though, the use of PCA improves the LAMAP algorithm's ability to discern potential differences between locations in the study area, which would make differentiating high- and low- potential areas easier. We suspect that this latter characteristic of PCA may also be a proxy for human cognitive processes that use observation of variability in the landscape as the basis for deciding where certain activities should take place.

## The Tanana Valley

The Tanana River is a tributary of the Yukon River and lies between the Alaska Range to the south and the Yukon River to the north. It is fed by meltwater from glaciers in the Alaska Range and flows northward to the Tanana Lowlands. The bottom of the Tanana Valley varies in width from less than a kilometer to 6.4 kms [26]. The part of the valley on which the present study focused comprises rounded ridges and bedrock hills, with a maximum elevation of 681 meters [27]. Large, free-standing craggy rock outcrops, known as tors, are found on summits or ridge crests. The valley floor is covered in thick layers of river silt and lowland loess [26]. Strong katabatic winds (high-density dry air) coming off the Alaska Range have been active in the area for millennia, especially during the Late Pleistocene, and are responsible for substantial loess deposits. The thickest concentration of loess soils, made up of minerals from igneous and metamorphic rock, are found on lower north-facing slopes.

The average summer temperature in the valley is 17 degrees Celsius and the average winter temperature is -23 degrees Celsius. The typical annual precipitation in the valley is 27 centimeters [28]. The area is considered a 'Dfc climate type', which is a subarctic climate [29, 30].

Today, the Tanana Valley is covered by subarctic boreal forest of the Nearctic Ecozone [28], dominated by black spruce (*Picea mariana*), followed by birch (*Betula* sp.), balsam poplar (*Populus balsamifera*), white spruce (*Picea glauca*), tamarack (*Larix laricina*), and aspen (*Populus* sp.). Common shrub species include lingonberry (*Vaccinium vitis-idaea*), birch, alder (*Alnus* sp.), Labrador tea (*Rhododendron tomentosum*), prickly rose (*Rosa acicularis*), and blueberry (*Vaccinium* sp.) [31]. Animal species include moose (*Alces alces*), caribou (*Rangifer tarandus*), Dall sheep (*Ovis dalli*), black and grizzly bears (*Ursus americanus* and *U. horribilis*), upland birds and waterfowl, and freshwater and anadromous fish.

The Tanana Valley has been inhabited by humans for at least 14,500 years [32, 33]. Earliest occupation occurred when the landscape was dominated by shrub tundra and continued into the Younger Dryas (c. 13,000 to 12,000 cal BP) when colder conditions created herb tundra. After 12,000 cal BP the climate became warmer and wetter, and tundra was replaced by boreal forest. Analyses of pollen from the site of Broken Mammoth indicate that the transition from steppe to boreal forest was complete by around 11,000 cal BP [32]. There have only been minor environmental changes in the valley in the intervening period [34].

The pre-colonial history of the Tanana Valley is currently divided into four major time periods [35]. The first is the Beringian Tradition (or Eastern Beringian Tradition), which dates from the initial occupation at 14,500 cal BP to around 12,500 cal BP [32]. A variety of lithic and bone technologies are known from this period, some of which appear to derive from ancestral cultures in western Beringia. Subsistence was based primarily on hunting large game, notably bison and wapiti (*Cervus elaphus*), with other large ungulates, and some smaller mammals and birds represented less frequently in faunal assemblages [36].

At about 12, 500 cal BP the Beringian Tradition was replaced by the American Paleoarctic Tradition (also known as the Denali Culture) [34, 35]. Although there was considerable continuity in subsistence patterns, bison and wapiti were not as common as in the previous period, probably reflecting the establishment of boreal forest [34, 35]. The American Paleoarctic Tradition extended to about 7000 cal BP.

The Northern Archaic Tradition was widespread in Alaska and Yukon from around 7000 cal BP [37]. Although there was some continuity with the preceding American Paleoarctic Tradition in lithic technology, a number of distinctive new tool types appeared, such as notched projectile points and microblades produced with different techniques. Subsistence patterns also changed, with a greater emphasis on caribou hunting, and a marked decline in bison and wapiti. Smaller mammals, such as hare (*Lepus americanus*), beaver (*Castor canadensis*), canids, and bear were hunted more frequently [36]. This appears to relate to the widespread establishment of boreal forest in the lowlands, and a greater use of upland environments where caribou could be intercepted. Although fish had been exploited since late Pleistocene times [33], fishing became more common during this period [36].

A significant change in settlement and subsistence occurred with the emergence of the Athabaskan Tradition between 1700 cal BP and 1000 cal BP [35, 36]. Better preservation of organic artifacts in more recent sites may exaggerate the technological changes, but the lithics clearly demonstrate a shift from osseous composite projectile points (atlatl darts) to bow and arrow. Winter villages, including semi-permanent pithouse structures and storage features, were located along salmon rivers. Caribou and moose became even more important big game animals, and fur-bearing animals and fish became more common in faunal assemblages [36].

These patterns continued into the colonial period. At the time of contact with Europeans, the Tanana Valley was occupied by Athabaskan-speaking groups [35]. The name 'Tanana' comes from the Denaakke term 'tene no, tenene', which means 'trail river' [38]. Tanana Athabaskans were partially sedentary; their semi-permanent settlements were located in the valley's lowland [39]. They hunted big game and smaller fur-bearing species. They also fished for salmon and trout and collected berries and other plant resources. Today, the economy of Tanana Athabaskans is a mixed cash-subsistence one, like many other Indigenous communities in Alaska, and still includes the hunting and gathering of wild resources [39, 40].

## Materials and methods

Six Digital Elevation Models (DEMS) for the study area were obtained from the US Geological Survey's publicly-accessible National Elevation Dataset [41]. These DEMs were imported and

merged into a single geo-referenced raster. The DEMs acquired were very high resolution, with each raster cell measuring 5 x 5 m. This is unusually fine-grained for publicly-accessible satellite data. Typically, the average resolution for such data is 30 x 30 m. The higher 5 x 5 m resolution had the potential to produce high-precision estimates, but it had implications for processing the data. In their raw form, the six DEMS included 1.6 billion 5 x 5 m cells, which would have meant unreasonably long processing times, even with a high-performance computing cluster. To reduce the amount of processing time, the six DEMS were resampled at 15 m x 15 m resolution by using a moving-average window after they were stitched together. The lower-resolution raster map was then cropped to fit the study area. Lastly, the map was re-projected from a geographic to Universal Transverse Mercator (UTM) projection [42], with the longitude and latitude coordinates converted into meters.

We collected information about the location and age of 182 archaeological sites in the middle Tanana River valley from the Alaska Heritage Resources Survey, with permission from the State of Alaska's Office of History and Archaeology (S1 Table). Twelve of the sites predate 10,000 cal BP; 17 date between 10,000 cal BP and 5,000 cal BP; 23 sites post-date 5,000 cal BP; and 130 are currently of unknown age. The sites' age estimates are based either on radiocarbon dates from excavated material or stylistic characteristics of formal artifacts, mainly flaked stone projectile points.

We used six landscape variables to characterise the CSAs around archaeological sites: 1) elevation; 2) slope; 3) aspect; 4) distance to drainage; 5) cumulative viewshed; and 6) convexity. These variables were selected for three main reasons. First, there is reason to think that they would have been important in relation to hunter-gatherers' use of the valley. Second, they are clearly related to other environmental variables that would have been important for hunter-gatherers, such as vegetation, temperature, and soil. Third, they represent landscape features that have likely changed little over time and are readily available from satellite imaging data. It is important to note that we did not use vegetation cover as a variable because palaeoecological records available for the area do not allow us to determine local vegetation at a scale that could be applied to site locations or relatively small raster cells.

The variable 'elevation' was obtained as raster digital elevation models (DEMs) created by NASA from satellite images [41]. These files were provided ready-to-use with the necessary geographic information. The remaining variables were all derived from the DEMs using algorithms available in GRASS [43]. The first were 'slope' and 'aspect', which were then used to extract drainages with another GRASS algorithm. After extracting the drainages, a raster map containing the distance from every cell in the study area to the nearest major drainage cell was extracted from the drainages map. The variable 'convexity' describes the degree to which a cell represents a convex/concave location on the landscape, i.e. the degree to which the cell is on a hill-like surface or a depression-like surface. It was derived from the elevation data using the GIS spatial algorithm 'Terrain Surface Convexity' provided through SAGA (System for Automated Geoscientific Analyses) [44]. The variable 'cumulative viewshed' describes the visibility of a given cell in the study area from a large grid of points that was overlaid on the area using QGIS's 'Viewshed Analysis' algorithm. The grid contained thousands of points spaced 1,000 m apart covering the entire study region. Values for 'cumulative viewshed' were derived from the elevation data, assuming a tress-less, obstruction-free view of the land surface from a height of 1.6 m.

Every cell in the study area was compared to those cells in a 1 km diameter CSA area around each known-site. The size was chosen to be consistent with previous LAMAP case studies [16, 19]. However, we also think that it represents a reasonable scale of landscape decision-making for hunter-gatherers engaged in a range of potential activities. There were roughly 3,500 15 m x 15 m cells in each 1 km diameter sample area, depending on where precisely the circle overlapped the cells. The simplest way to quantify the character of the landscape around a known-

site was to sample places in the vicinity of that site, measure a given landscape variable, and estimate the corresponding frequency distribution of measurements. From this distribution it was possible to estimate the empirical probability of finding a spot within the site's CSA that is similar to a target cell not within the CSA of a known-site. Each target cell was then assigned a probability value that reflects its similarity to the landscape around known sites. The procedure was repeated for each cell and for each of the six landscape variables. Once these calculations had been completed, all target cells were divided into five classes (quintiles) of equal size based on their probability values and ranked from 1 (lowest potential) to 5 (highest potential). The distribution of target cells in each class was then mapped.

We carried out two LAMAP analyses. In the first, 90 sites were randomly selected to build a LAMAP model of archaeological potential. Once the model was developed, the remaining 92 sites were used to test the model. We reasoned that, if the model were valid, we should see more of second set of known-sites occurring in areas that had been identified as high potential, and fewer known-sites in low potential areas.

In the second analysis, we used the 12 pre-10,000 cal BP sites to create a LAMAP model and 60 post-10,000 cal BP known-sites to test it. Only sites that have been assigned to a time period were employed in this analysis. In addition, the post-10,000 cal BP set of sites did not include any multi-component sites with a pre-10,000 cal BP component. This means that sites were not used to both train and test the model.

We used regression analysis to assess the performance of the LAMAP models. The LAMAP values were binned into five ordinal classes using quintiles. We then examined the distribution of validation sites, according to the LAMAP class of the raster cell in which they were located. Following the approach to validation taken in Carleton et al.'s [18] Belize case study, we counted the number of validation sites located in cells of each of the five classes and then compared the number of sites to the corresponding predictive class value. The comparison was carried out with a simple Poisson regression model, which was chosen because the data consist of counts of sites per predictive class. In each model, we used LAMAP class as the 'predictor/independent' variable and validation site count as the 'response/dependent' variable. We then examined the estimated regression coefficients. We reasoned that, if the LAMAP models had predictive power there should be a significant positive relationship between LAMAP class and site count, i.e., there should be a positive regression coefficient and a significant p-value. These model parameters were estimated using MCMC and a Bayesian regression framework. The MCMC simulations were run for a minimum of 20,000 iterations to ensure convergence. The priors used for the model parameters had a wide variance.

Most analyses were conducted in the R software environment [45] with some data processing involving Python (v.2.7) and QGIS (v.>3) http://qgis.osgeo.org). An R package (pre-release) for running LAMAP analyses can be downloaded from https://github.com/wccarleton/lamap. The LAMAP surfaces were computed on the WestGrid high performance computing cluster that is managed by Canada's national computing science partnership, Compute Canada (https://www.computecanada.ca/home/). Links to all scripts required to replicate the analyses are provided in the Supporting Information. The archaeological site data have also been included in a Supporting Information file.

## Results

### Cross-validation test of LAMAP

The results of the analysis involving the randomly selected training and validation subsamples were encouraging (Fig 2). Visual inspection shows that the validation sites used to test the

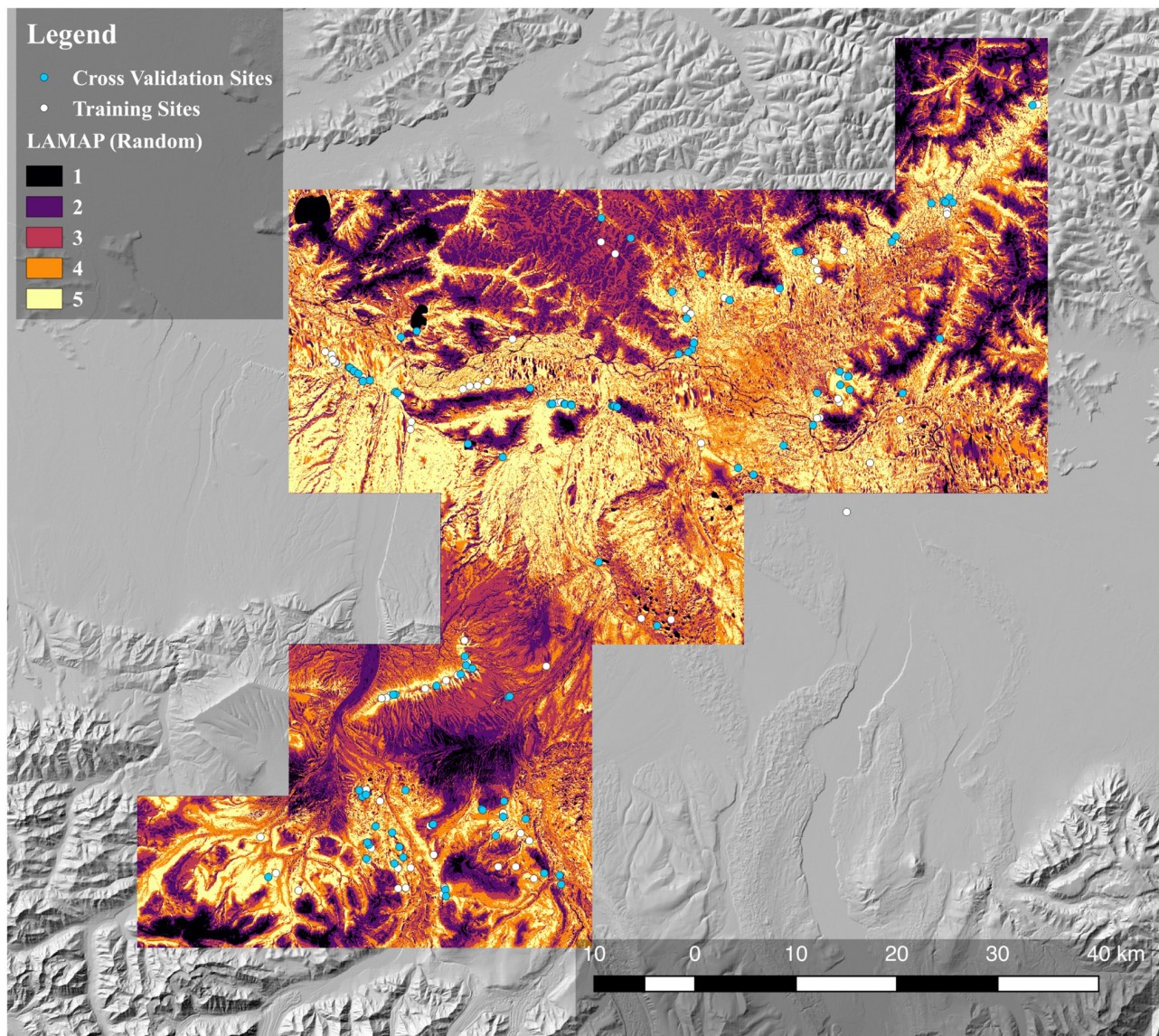

**Fig 2. Map of archaeological potential, randomly selected sites.** Archaeological potential was modeled on 90 randomly selected sites (white) and tested with 92 other sites (blue). The five classes of archaeological potential are coded from 1 (lowest potential) to 5 (highest potential).

model (blue circles) are generally located in regions predicted by the model to fall in a higher LAMAP class and therefore have higher archaeological potential.

This visual impression is supported by a bar-plot of validation site counts and LAMAP classes (Fig 3). As the plot shows, more of the sites used to test the LAMAP model (blue circles) are located in areas with LAMAP Class 3 or higher, while half of the validation sites are in Class 4 or Class 5 locations (see also Table 1). The differences between potential estimates across classes is not large unless one compares a low class, e.g. Class 1, to a high one, e.g. Class 5. There is also an apparent drop in the number of sites counted in Class 5 cells. Nevertheless, the regression results indicate that the higher classes yield more sites on average (Fig 3). The average size of the increase in site count associated with increases in LAMAP class helps to put the results into perspective. According to the regression model, increasing the LAMAP class

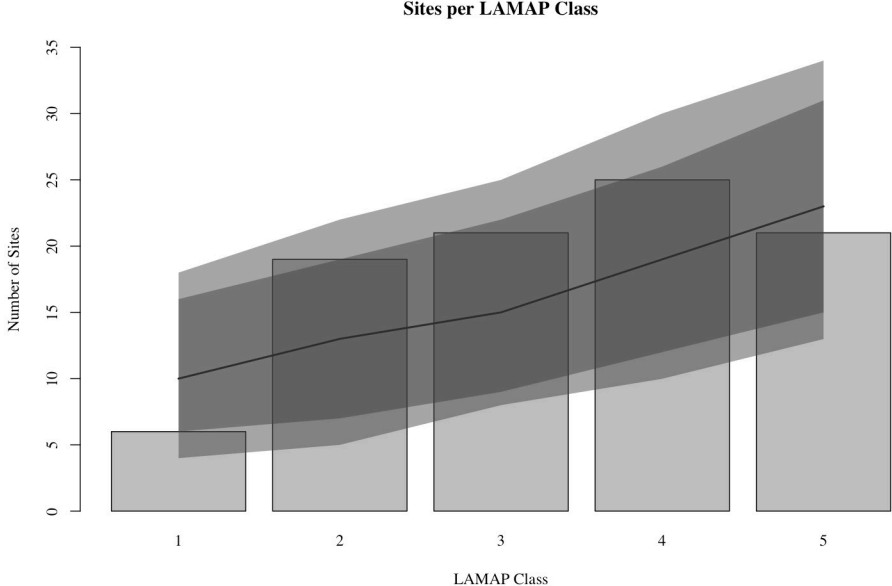

**Fig 3. Relationship between LAMAP class and the number of sites found on land of each class, randomly selected sites.** The grey vertical bars indicate the number of sites located in cells with the given potential estimate. The solid black trend-lines indicate the expected number of sites for a given LAMAP class according to the Poisson regression model. The 95% and 99% posterior predictive intervals for the regression models are indicated by the darker and lighter grey shaded ribbons, respectively.

by one level corresponds to a 12% increase in the number of sites identified, on average. Thus, going from a Class 1 region to a Class 5 region, we would expect to see an 88% increase in site counts.

## Pre/Post-10,000 cal BP test of LAMAP

The results of the analysis in which we compared pre- and post-10,000 BP sites were similarly encouraging. Most of the post-10,000 cal BP sites are located in areas designated Class 3 or higher, with most in Class 4 or 5 locations (Fig 4). This impression is confirmed numerically by the site counts per LAMAP class (Table 1). Thus, areas of high archaeological potential defined on the basis of Late Pleistocene sites also contained most of the later Holocene sites. Consistent with the previous set of results, the regression model indicates a significant positive association between validation site counts and LAMAP classes (Fig 5). The regression also

**Table 1. Number of sites per LAMAP class in the two Tanana Valley analyses.** 'Random' shows the results obtained when the model was built on 90 randomly selected sites and tested with a different set of 92 randomly selected sites. 'Pre/Post' shows the results obtained when the model was built on pre-10,000 cal BP sites and tested with post-10,000 cal BP sites.

| Class | Random | Pre/Post |
|---|---|---|
| **1** | 6 | 0 |
| **2** | 19 | 4 |
| **3** | 21 | 8 |
| **4** | 25 | 13 |
| **5** | 21 | 8 |
| **Total** | 92 | 33 |

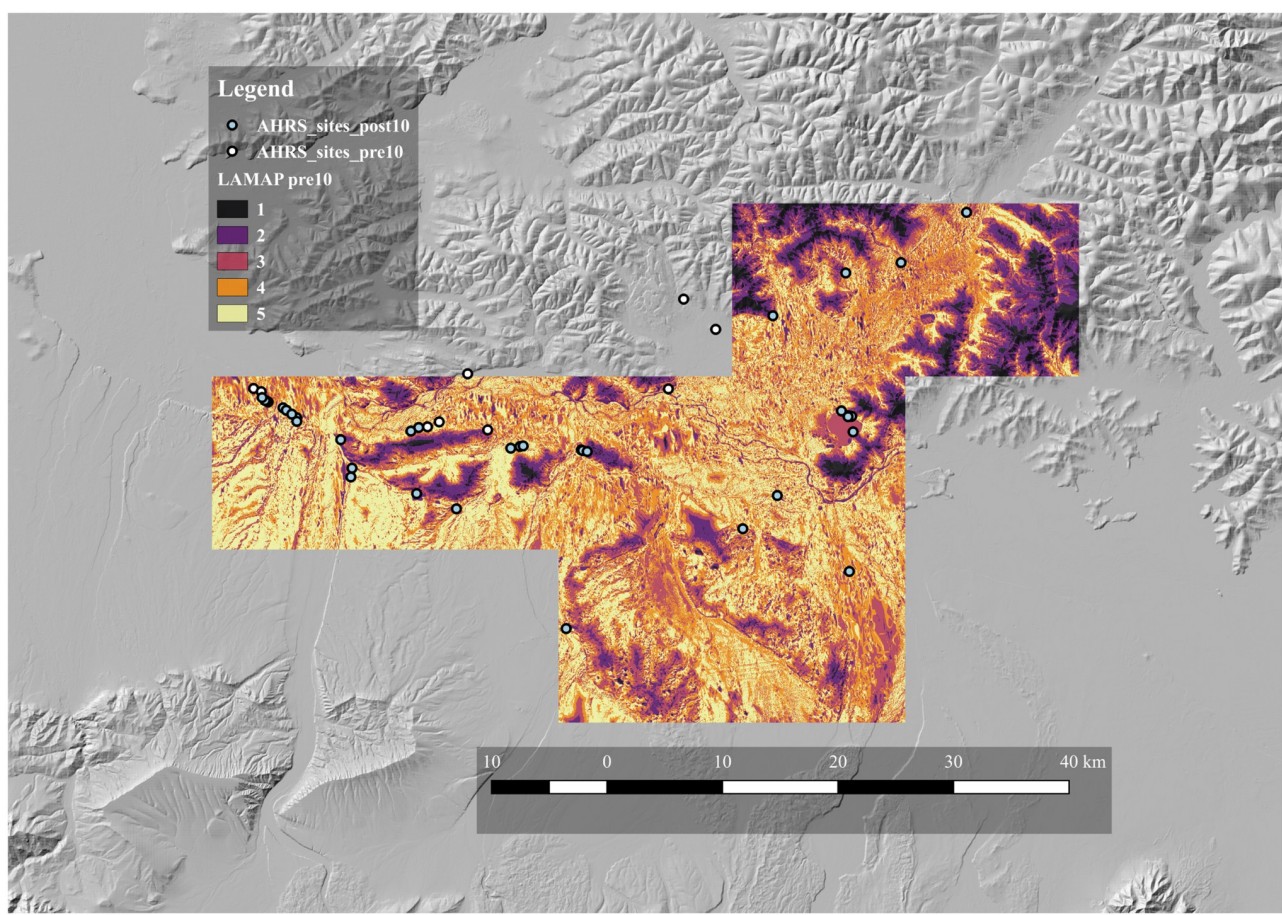

**Fig 4. Map of archaeological potential, pre/post-10,000 cal BP sites.** Archaeological potential was modeled on pre-10, 000 cal BP sites (white) and tested with post-10,000 cal BP sites (green). The five classes of archaeological potential are coded from 1 (lowest potential) to 5 (highest potential).

indicated that a one level increase in class corresponded to a 50% increase in the number of identified sites on average. Moving from a Class 1 to a Class 5 region would, therefore, be expected to yield over 650% more sites.

## Discussion and conclusions

The LAMAP method of estimating the archaeological potential of an area has previously been successfully tested in relation to archaeological sites associated with agricultural societies [18, 46]. In the present study, we evaluated the ability of LAMAP to estimate archaeological potential in an area occupied exclusively by hunter-gatherers.

The area on which the study focused, the Tanana Valley in central Alaska, was occupied exclusively by hunter-gatherer groups for more than 14,500 years. In view of the length of time the valley has been occupied, we used physiographic variables that probably persisted over long time periods such as elevation and slope to generate models of archaeological potential, rather than variables that are known to vary on short time scales, such as vegetation or distribution of game animals.

In the first test of LAMAP, we used the location of a random selection of known sites from all time periods to create a model of archaeological potential for the study area. We then evaluated the model by assessing the counts of a second set of randomly selected known sites in

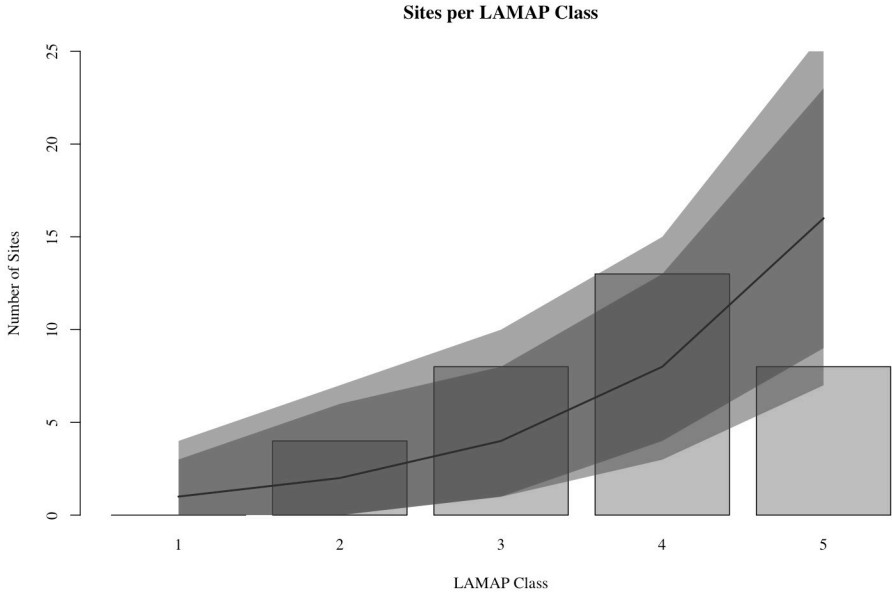

**Fig 5. Relationship between LAMAP class and the number of sites found on land of each class, pre/post-10, 000 cal BP sites.** See Fig 3 for key.

relation to the mapped archaeological potential. In the second test, we built a LAMAP model from 12 known sites that pre-10,000 cal BP. This model was then tested with known sites dated by radiocarbon dating or artifact typology to later than 10,000 cal BP. In both analyses, areas predicted to have higher archaeological potential contained higher frequencies of sites from the validation set of known sites. Thus, the present study indicates that LAMAP is useful for estimating archaeological potential in relation to hunter-gatherer sites.

LAMAP has now been tested four times, twice with data pertaining to archaeological sites produced by agricultural societies [18, 19], and twice with data for archaeological sites created by hunter-gatherer societies (this study). In each application LAMAP predicted areas with varying degrees of potential for discovery of archaeological sites, and in each case a validation process demonstrated that areas identified as higher potential did have more sites. However, LAMAP was not equally effective in the four tests. Carleton et al. [18] found that an increase of one in the LAMAP class (e.g. from Class 2 to Class 3) corresponded to an increase in site count of 32%, while an increase from the lowest class to the highest class corresponded to an increase in site count of 200%. The equivalent figures for the Wilett et al. study [19] were 27% and 161%, respectively. In the first of two tests reported here, an increase of one in the LAMAP class corresponded to an increase in site count of 12%, while an increase from the lowest class to the highest class corresponded to an increase in site count of 88%. In the second of tests we carried out with the Tanana Valley data, an increase of one in the LAMAP class corresponded to an increase in site count of 50%, while an increase from the lowest class to the highest class corresponded to an increase in site count of 657%. Thus, the best-performing model is over seven times more effective than the worst-performing one. Significantly for present purposes, there is not a difference in performance between the models based on sites produced by hunter-gatherer sites and models based on sites generated by agricultural societies. One of the hunter-gatherer models was less effective than either of the models for agricultural societies, but the other hunter-gatherer model performed better than both of the models for

agricultural societies. This is striking given that the hunter-gatherers in question were mobile and therefore did not create permanent structures.

A potential criticism of LAMAP is that research and sampling biases affecting the training data propagate into the model's potential estimates. For example, if known archaeological sites had been located by archaeologists who preferentially searched on certain landforms, the physiographic characteristics of those preferred landforms would be over-emphasised in the model. However, this is a problem with all predictive models, including those built upon random or stratified random sampling strategies, because those strategies are also constrained by differential survival of sites and our ability to detect sites that have survived.

We think predictive modelling is useful despite training data biases for two main reasons. First, predictions of archaeological potential are important for heritage resource management even if those predictions refer only to a subset of the total unknown archaeological record. For management purposes it is better to have an idea of where some sites are likely to be located than no idea at all. Flawed predictions have value so long as decision-makers are aware that novel site types (not represented in the available training data) may be located in low-potential zones, which means low-potential areas cannot be treated as if they are completely devoid of archaeological sites. Second, along similar lines, predictive models have scientific value so long as sampling limitations are recognised. We gain insight into past land-use behaviour even if only a subset of past behaviours are available to us.

A second methodological issue that requires comment is the size of the circular sampling area (CSA). The mental algorithm that past hunter-gatherers used to select a place for certain activities is likely to have been both complex and situational. We selected a 1 km diameter circle for this analysis to ensure comparability with results from previous applications of LAMAP, but future research could examine whether smaller or larger areas around a site provide better predictions about archaeological potential. The granularity of available environmental data also may have to be considered when selecting the diameter of the CSA.

We selected the Tanana Valley as a test case because of the lengthy history of hunter-gatherer land use and the relatively low degree of landscape modification and disturbance in recent centuries. We caution that further work would be required to refine the model of archaeological potential in the Tanana Valley for both heritage management and research purposes. However, in addition to demonstrating that LAMAP is useful for estimating archaeological potential in relation to hunter-gatherer sites, the study reported here provides some preliminary conclusions on hunter-gatherer site location preferences in central Alaska that could be refined through further study. The first LAMAP model indicated that the Tanana Valley's prehistoric inhabitants favoured locations that were relatively close to water but also elevated above the surrounding landscape, while the second LAMAP model suggested that this preference was not greatly affected by the changes in ecology associated with the transition from the Pleistocene to the Holocene. These findings are consistent with the conclusions that a number of archaeologists have reached on the basis of fieldwork. For example, Aigner and Gannon [46] reported the results of archaeological surveys of pipeline routes in central Alaska and noted that high potential areas included locations close to fresh water and promontories with good visibility. A useful next step would be to investigate why such locations were preferred. The obvious explanation for favouring close proximity to water is that it offered access to aquatic and semi-aquatic food resources, but there are other possibilities. For example, the preference could be associated with the greater ease of travel along frozen waterways in winter. Similarly, there are several potential reasons why elevated areas were selected. For instance, they could have enabled prey to be spotted more easily. Alternatively, they may have been favoured because they were drier and therefore better for camping.

There are some other possibilities for archaeological research in the Tanana Valley that arise from our study. One is that the two LAMAP models could be field tested through archaeological surveys designed to assess sections of the landscape that are designated by LAMAP as having different degrees of archaeological potential. The two LAMAP models could also be used to direct future research and heritage management in central Alaska. Because this is a key area for understanding the early peopling of the Americas, LAMAP could also be used to direct archaeologists to locations that have high potential for finding early sites. Lastly, in future management projects LAMAP could be used to identify areas in central Alaska that require more intensive survey or areas that should be avoided by high-impact development activities.

The results of the present study also support the use of LAMAP in the search for hunter-gatherer sites in other parts of the world. Such sites are not usually found through remote sensing. Rather, their discovery relies on surface examination and subsurface testing, both of which are labour-intensive and time-consuming. There are large areas of the world (e.g. most of Australia and much of North America) where the archaeological record was exclusively the product of hunter-gatherers until the last few centuries, and in many of these regions more sites are required to better understand the diversity of cultures in time and space. As a tool for identifying areas of high archaeological potential, LAMAP can focus attention on parts of the landscape that are likely to be most productive for future research.

Given the strong correlation between site location and physiographic features revealed by the two LAMAP models, it may also be possible to extend the use of LAMAP to areas that contain no known hunter-gatherer sites, and where exploration is likely to be expensive and difficult. For example, post-Pleistocene sea level rise has inundated areas that were formerly inhabited by hunter-gatherers. Some of these areas may be critical for establishing important events and processes in our past, such as the colonisation of North America's Pacific coastal plain by PaleoAmericans, or the utilisation of Doggerland in what is now the North Sea by hunter-gatherers during the early Holocene. In such situations it may be possible to model hunter-gatherer site location characteristics from nearby landscapes and then apply the model to reconstructed terrestrial landscapes that are now submerged. Locations with high archaeological potential could then be tested using underwater sampling techniques such as coring or remote sensing.

In addition to its use as a research tool, LAMAP could also be used to build models of archaeological potential that can be applied in conservation archaeology, where the goal is to either maximise the number of sites discovered, or identify areas that should be avoided by ground-altering development processes, such as the construction of roads and pipelines, or open-pit mining. This application would be useful in regions where large areas of land have yet to be explored and where hunter-gatherer sites are difficult to detect except through intensive foot survey and subsurface testing, such as boreal forest regions of Canada or the deserts and xeric shrublands of Australia.

## Supporting information

**S1 Table. AHRS sites database.** Data from the Alaska Heritage Resources Survey, the State of Alaska's Office of History and Archaeology. These sites were used to build and test LAMAP models in the Tanana valley. Locational information for each site was made available for the purposes of research, but locational data are redacted here in order to protect heritage resources.
(DOCX)

**S2 Table. Links to scripts.** These links provide access to scripts used in the LAMAP analysis.
(DOCX)

## Acknowledgments

We are grateful to Charles Holmes, Gerad Smith, and Diane Hanson of the University of Alaska for their assistance with this project, which included hosting a visit to the Tanana Valley by RR.

## Author Contributions

**Conceptualization:** Rob Rondeau, W. Christopher Carleton, Mark Collard, Jonathan Driver.

**Data curation:** W. Christopher Carleton.

**Formal analysis:** W. Christopher Carleton.

**Funding acquisition:** Rob Rondeau, Mark Collard.

**Investigation:** Rob Rondeau.

**Methodology:** Rob Rondeau, W. Christopher Carleton, Mark Collard, Jonathan Driver.

**Project administration:** Rob Rondeau.

**Resources:** Rob Rondeau, W. Christopher Carleton.

**Software:** W. Christopher Carleton.

**Supervision:** Mark Collard, Jonathan Driver.

**Validation:** Rob Rondeau, W. Christopher Carleton.

**Visualization:** W. Christopher Carleton.

**Writing – original draft:** Rob Rondeau.

**Writing – review & editing:** Mark Collard, Jonathan Driver.

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
