## [Decision Letter · Decision Letter 0]

6 Oct 2021

PONE-D-21-23001Testing the locally-adaptive model of archaeological potential with hunter-gatherer sites in the Tanana River valley, AlaskaPLOS ONE

Dear Dr. Driver,

Thank you for submitting your manuscript to PLOS ONE. After careful consideration, we feel that it has merit but does not fully meet PLOS ONE’s publication criteria as it currently stands. Therefore, we invite you to submit a revised version of the manuscript that addresses the points raised during the review process.

Dear authors,

Following two differing opinions from reviewers, and my own assessment of the manuscript submitted to PLOS One, I am recommending Acceptance of your submission subject to major revision.

Please find below detailed comments from both reviewers. As highlighted by R1, I found your approach with the improved modelling informative, particularly with regard to overcoming aspects of diversity in site location and landscape/environment related variables. Furthermore, I laud the efforts and approach of the authors in using and building on open source software. At the same time, as raised by R1, it would be good to state more explicitly how analysis scripts will be made available, and if the fully open access deposition of these is not possible (e.g., due to permit, government regulation, legal requirement, etc. related restrictions) please consider if protected deposit at a research data repository could be possible. In your revision, please ensure that all data sources, analysis components, scripts, and details of access to other sources of information are either included in supplementary materials and/or clearly linked to in existing data repositories.

Most importantly, Reviewer 2 has serious concerns regarding the reproducibility of the analyses presented. I am not convinced that a wider (i.e., by the commercial archaeology sector) lack of access to high-power computing facilities has a bearing on the reproducibility of this manuscript. At the same time, the responsibility falls on the authors to make explicit all the necessary hardware, software and datasets to ensure that this novel and important methodology can be applied as widely as possible. To this effect, please note that PLOS One is very generous on word count/space made available for major research articles, and most types of data/scripts can be easily included as supplementary materials. Should you have any further queries please do not hesitate to get in touch with myself or with the editorial office.

We look forward to receiving your revised manuscript.

Kind regards,

Ceren Kabukcu, PhD

Academic Editor

PLOS ONE

2. Please consider in addition whether it meets PLOS ONE criteria for papers that describe new methods or software for applications. Specifically, these reports must meet the criteria of utility, validation, and availability, which are described in detail at http://journals.plos.org/plosone/s/submission-guidelines#loc-methods-software-databases-and-tools.

3. We note that Figures 1,2 and 4 in your submission contain [map/satellite] images which may be copyrighted. All PLOS content is published under the Creative Commons Attribution License (CC BY 4.0), which means that the manuscript, images, and Supporting Information files will be freely available online, and any third party is permitted to access, download, copy, distribute, and use these materials in any way, even commercially, with proper attribution. For these reasons, we cannot publish previously copyrighted maps or satellite images created using proprietary data, such as Google software (Google Maps, Street View, and Earth). For more information, see our copyright guidelines: http://journals.plos.org/plosone/s/licenses-and-copyright.

a. You may seek permission from the original copyright holder of Figures 1, 2 and 4 to publish the content specifically under the CC BY 4.0 license. 

Additional Editor Comments (if provided):

Reviewers' comments:

Reviewer's Responses to Questions

**Comments to the Author**

1. Is the manuscript technically sound, and do the data support the conclusions?

Reviewer #1: Yes

Reviewer #2: Yes

2. Has the statistical analysis been performed appropriately and rigorously? 

Reviewer #1: Yes

Reviewer #2: Yes

3. Have the authors made all data underlying the findings in their manuscript fully available?

Reviewer #1: Yes

Reviewer #2: Yes

4. Is the manuscript presented in an intelligible fashion and written in standard English?

Reviewer #1: Yes

Reviewer #2: Yes

5. Review Comments to the Author

Reviewer #1: The LAMAP approach certainly overcomes some issues of predictive modelling and in particular focussing on variability of landscape parameters and not using non-site locations. Further more, an application to hunter-gatherers is a welcome addition to the previous papers on the topic.

Though, this is an excellent paper, the authors might want further improving it by addressing some points:

- In the introduction reasons for predictive modelling are mentioned. A well known problem is that this approach can enable us to find some sites and prevents us from finding other sites because the research focus bias. The other sites might be more telling. The better the prediction methods, the higher the bias. A sentence why this approach is nevertheless useful and valuable would be nice.

- It would be helpful to say something about the size of the surrounding of the sites in the method section. Later it is mentioned that the size is 1 km diameter circular quadrat (my English is not good enough to understand this term. How can a quadrat be circular? Four corners are a quite bad approximation of a circle), but this decision is not explained. A connection to site catchment approaches could make sense. Anyway, why this particular size?

- The PCA is another important issue. The benefits are explained but it remains open at which level the new variables are skipped and why ignoring this information would be appropriate? Since PCA is ordering the new variables according to variability, an overvaluation of large variability is inevitable. It should be explained how this concept is supported by the underlying theory of the LAMAP approach.

- Finally, the analysis uses open source software which is very good. But is the analysis reproducible? The data will fully available but are the analysis scripts also available? (It might be mentioned that the reviewer does not consider "accessible to qualified researchers" as fully available and does not belief that "protect sites from vandalism" is an appropriate reason but that need not be the subject of this report, as this is not always a decision of the authors.

Feel free to contact me if you have any questions.

Oliver Nakoinz

Reviewer #2: Summary

This paper employs an off-the-shelf, but relatively new, method for generating statistical predictive models for archaeological site locations and does so to evaluate its efficacy for use by hunter-gatherer archaeologists. It is cleanly written and well organized, although the latter could be improved a touch to strengthen the manuscript. I also found the methodology appropriate and results supported by the data. The conclusions make sense too. While a generally sound paper, given that it neither presents a novel method nor evaluates a well-developed archaeological question, I feel it a better candidate for a regional journal. I also have issues with replicability that I suggest the authors consider and these are discussed below.

Comments

1. Would like to see the hypothesis in play stated explicitly along with the predictions under test.

2. This paper seems of two minds. I found it unclear if this is an evaluation of the methodology or the exploration of an archaeological question. I would pick one and concentrate on making the relevant point. Because this in not a test of a novel methodology, I might focus instead on using it to answer a well developed and (at least) regionally important research question about settlement patterns in the Tanana River Valley. As is, we get, did site locations change through time? as the research question, but why this is an important question is never discussed in any meaningful way.

3. While I think that points one and two can be easily addressed, I do have issues with the replicability of this study and this is one of the important publication criteria for PLOS. The primary issue here is that very few of us have access to national laboratory level main frame computing clusters. My concerns here are twofold. First is the observation that to replicate this study one needs access to similar computing power. This limits replicative studies to only those with similar computing capabilities and renders this method an “insiders” approach. Second, this study appears aimed at cultural resource management archaeologists, the vast majority of which are working on Best Buy quality and performance-level laptop or desktop computers. To me, this is a substantial and important disconnect between the study and suggested real-world application – I can‘t ready replicate the study, nor can I (or the vast majority of archaeologists, especially those in the target audience) employ the methodology for my own research.

6. PLOS authors have the option to publish the peer review history of their article (what does this mean?). If published, this will include your full peer review and any attached files.

Reviewer #1: No

Reviewer #2: No

---

## [Author Response · Author response to Decision Letter 0]

2 Mar 2022

Responses to comments on PONE-D-21-23001

Editor’s comments

Editor comment: At the same time, as raised by R1, it would be good to state more explicitly how analysis scripts will be made available, and if the fully open access deposition of these is not possible (e.g., due to permit, government regulation, legal requirement, etc. related restrictions) please consider if protected deposit at a research data repository could be possible. In your revision, please ensure that all data sources, analysis components, scripts, and details of access to other sources of information are either included in supplementary materials and/or clearly linked to in existing data repositories.

Response: We address the issue of protecting the data on the locations of the archaeological sites in our response to Reviewer #1 (see below). Regarding scripts, examples of R and Python scripts can be found in a GitHub repository (https://github.com/wccarleton/lamap_tanana) that will be archived with Zenodo following acceptance of the manuscript. We will provide the link to this repository in the Supporting Information.

Editor comment: Most importantly, Reviewer 2 has serious concerns regarding the reproducibility of the analyses presented. I am not convinced that a wider (i.e., by the commercial archaeology sector) lack of access to high-power computing facilities has a bearing on the reproducibility of this manuscript. At the same time, the responsibility falls on the authors to make explicit all the necessary hardware, software and datasets to ensure that this novel and important methodology can be applied as widely as possible. To this effect, please note that PLOS One is very generous on word count/space made available for major research articles, and most types of data/scripts can be easily included as supplementary materials. Should you have any further queries please do not hesitate to get in touch with myself or with the editorial office.

Response: We have addressed the concerns of Reviewer #2 about availability of computing resources below. We have included information about the sites we used in our study in the Supporting Information, and any bona fide researcher can access this information via a request to the State of Alaska Office of History and Archaeology. As noted above, the scripts and software will all be made available online in an open software repository (https://github.com/wccarleton/lamap_tanana). These include examples of the scripts used to run the analyses described in the paper and a pre-release R package for running LAMAP analyses more generally. The repository will be submitted to the Zenodo archiving service to be preserved in its current state in perpetuity.

Response: we believe that the Supporting Information with links to relevant scripts covers the issue of laboratory protocols and that no further action is required.

3. We note that Figures 1,2 and 4 in your submission contain [map/satellite] images which may be copyrighted. All PLOS content is published under the Creative Commons Attribution License (CC BY 4.0), which means that the manuscript, images, and Supporting Information files will be freely available online, and any third party is permitted to access, download, copy, distribute, and use these materials in any way, even commercially, with proper attribution. For these reasons, we cannot publish previously copyrighted maps or satellite images created using proprietary data, such as Google software (Google Maps, Street View, and Earth). For more information, see our copyright guidelines: http://journals.plos.org/plosone/s/licenses-and-copyright.

Response: Figures 1, 2 and 4 do not contain copyright materials. All maps were created using publicly available data. This is now reflected in figure captions or the text.

Reviewer #1

Reviewer #1 comment: In the introduction reasons for predictive modelling are mentioned. A well known problem is that this approach can enable us to find some sites and prevents us from finding other sites because the research focus bias. The other sites might be more telling. The better the prediction methods, the higher the bias. A sentence why this approach is nevertheless useful and valuable would be nice.

Response: The bias is introduced by sampling/research bias in the training data. It’s not clear why the bias would be worse given a method with more predictive power alone because better prediction doesn’t translate into greater bias. Any change in the degree of bias depends on prediction variance—i.e., if a method has a lower prediction variance, it might more frequently miss sites with characteristics not represented in the training data than another method that has a higher predictive variance (like the difference between estimates with small and large confidence intervals). At the moment, though, we don’t know what the LAMAP predictive variance is compared to other approaches—for that matter, we aren’t aware of any papers quantifying prediction variance for predictive modelling in an archaeological context at all. We have added a section on bias in the Discussion.

Predictive modelling is useful despite training data biases for two main reasons. First, accurate predictions of archaeological potential are important for heritage resource management even if we suspect those predictions refer only to a subset of the total unknown archaeological record—it is better to at least have a sense of where some sites are likely to be located. The predictions continue to have value so long as decision-makers are aware that novel site types (not represented in the available training data) may be located in low-potential zones, which means low-potential areas cannot be treated naively as if they really are devoid of archaeological material. Second, along similar lines, predictive models, and land-use models more generally, have scientific value so long as sampling limitations are recognized. We gain insight into past land-use behaviour even if only a subset of past behaviours are available to us.

Reviewer #1 comment: It would be helpful to say something about the size of the surrounding of the sites in the method section. Later it is mentioned that the size is 1 km diameter circular quadrat (my English is not good enough to understand this term. How can a quadrat be circular? Four corners are a quite bad approximation of a circle), but this decision is not explained. A connection to site catchment approaches could make sense. Anyway, why this particular size?

Response: The awkward terminology was an editorial compromise. In previous papers on this topic we have used the term “catchment” only to be criticized by reviewers who understood the term to refer specifically to the source areas for the accumulation of a given material in a sink, like pollen, water, or in archaeological contexts resources gathered by humans—as in “catchment analysis”. We have changed the term to “circular sample area (CSA)” throughout the paper.

The size of the CSAs was a compromise. The diameter of the sampling areas is really a variable that could and should be explored further. We selected the size of 1km diameter because that was the size used in previous LAMAP studies and we wanted to be consistent so that the results would be comparable. We have added a section explaining the 1km diameter, and we have also added reference to the need for further research on this in the Discussion.

Reviewer #1 comment: The PCA is another important issue. The benefits are explained but it remains open at which level the new variables are skipped and why ignoring this information would be appropriate? Since PCA is ordering the new variables according to variability, an overvaluation of large variability is inevitable. It should be explained how this concept is supported by the underlying theory of the LAMAP approach.

Response: The decision about the number of PCs to retain during dimensional reduction always involves a trade-off. In our case, the trade-off balanced including more variables with diminishing discrimination potential against computational time. So, to some extent the selection of PCs was motivated by trying to maximize the discrimination between locations (on the basis of the variables measured) while minimizing computation time. Additionally, PCA is useful for discrimination precisely because it emphasizes variability. Observations may be close to one another quantitatively on more than one dimension (e.g., elevation, distance to water) independently, but at the same time far apart when correlations between those dimensions are considered. Using PCs instead of raw variables minimizes information redundancy and maximizes differences among observations making it easier to determine how similar one or more locations are with respect to all measured variables at once. Lower-ranked (low variability, low eigenvalue) PCs contain less discriminating information by definition (observed sites would all have similar scores) and, so, including them would have a lower impact on predictive estimates. Our choice to retain PCs involved considering eigenvalues (those above 1, which is a commonly used benchmark) and percentage of variance accounted for by all retained PCs.

That said, the reviewer has raised an interesting point about the potential connection between the PCA transformation and the actual land-use decision making process LAMAP is attempting to capture. Lower ranked PCs would, as we just explained, reflect combinations of variables that function poorly as discriminators between landscape locations. They contain information that cannot be used to tell one area from another. As a result, that information would be of little use to a human trying to distinguish between potential activity locations and, so, would have been much less consequential for deciding where to locate oneself. It seems much more likely that humans focus on characteristics that afford the greatest potential for discrimination between locations on the landscape, much like a PCA does. We have added some text that covers these points.

Reviewer #1 comment: Finally, the analysis uses open source software which is very good. But is the analysis reproducible? The data will fully available but are the analysis scripts also available? (It might be mentioned that the reviewer does not consider “accessible to qualified researchers” as fully available and does not belief that “protect sites from vandalism” is an appropriate reason but that need not be the subject of this report, as this is not always a decision of the authors.

Response: Links to examples of R and Python scripts will be included as Supporting Information and made publicly available online in a GitHub repository, as explained above. We were allowed to use data on the location of archaeological sites in Alaska on the understanding that detailed locational information would not be made public. This is standard practice with regard to the location of archaeological sites in North America and should not require further comment. Legitimate researchers can always obtain the data from the relevant authorities. We are guided by the following Society for American Archaeology statement: “An interest in preserving and protecting in situ archaeological sites must be taken in to account when publishing and distributing information about their nature and location” (Principles of Archaeological Ethics at https://documents.saa.org/container/docs/default-source/doc-careerpractice/saa_ethics.pdf?sfvrsn=75f1b83b_4). Therefore, we will not provide detailed information on site locations

Reviewer #2

Reviewer #2 comment: Would like to see the hypothesis in play stated explicitly along with the predictions under test.

Response: It’s not clear if the reviewer means a hypothesis about Tanana Valley prehistory, or a hypothesis about the application of LAMAP to hunter-gatherer sites. As noted in our response to the next comment (see below) the focus of the paper is on the method, rather than regional prehistory. We have made the methodological focus of the paper more explicit in the Introduction, and changed the title of the paper to emphasize that we are testing a method.

Reviewer #2 comment: This paper seems of two minds. I found it unclear if this is an evaluation of the methodology or the exploration of an archaeological question. I would pick one and concentrate on making the relevant point. Because this in not a test of a novel methodology, I might focus instead on using it to answer a well developed and (at least) regionally important research question about settlement patterns in the Tanana River Valley. As is, we get, did site locations change through time? as the research question, but why this is an important question is never discussed in any meaningful way.

Response: the reviewer has identified the need to make the primary focus of the paper more obvious. Our research did reveal some results that are relevant to Tanana Valley archaeology, but the goal of the study was to determine whether LAMAP can be used to assess archaeological potential of landscapes inhabited by mobile hunter-gatherers. We have rewritten parts of the Introduction to make this clearer and also changed the title of the paper. In addition, we have stated in the Discussion that further work would have to be done to support our preliminary conclusions about Tanana Valley prehistory.

Reviewer #2 comment: While I think that points one and two can be easily addressed, I do have issues with the replicability of this study and this is one of the important publication criteria for PLOS. The primary issue here is that very few of us have access to national laboratory level main frame computing clusters. My concerns here are twofold. First is the observation that to replicate this study one needs access to similar computing power. This limits replicative studies to only those with similar computing capabilities and renders this method an “insiders” approach. Second, this study appears aimed at cultural resource management archaeologists, the vast majority of which are working on Best Buy quality and performance-level laptop or desktop computers. To me, this is a substantial and important disconnect between the study and suggested real-world application – I can‘t ready replicate the study, nor can I (or the vast majority of archaeologists, especially those in the target audience) employ the methodology for my own research.

Response: The fact that access to national level computing resources is required for LAMAP is not a legitimate reason for discounting the research. Archaeologists with access to such resources include those at universities and in the public sector. For example, all academics in Canada can obtain access to the cluster we used, without charge. It is not unusual for academic archaeologists to collaborate with heritage managers and consultants, either through contract work or as research partners. We do not see the need to mention this issue in the manuscript.

---

## [Editor Report · Decision Letter 1]

7 Mar 2022

Does the Locally-Adaptive Model of Archaeological Potential (LAMAP) work for hunter-gatherer sites? A test using data from

the Tanana Valley, Alaska

PONE-D-21-23001R1

Dear Dr. Driver,

We’re pleased to inform you that your manuscript has been judged scientifically suitable for publication and will be formally accepted for publication once it meets all outstanding technical requirements.

Kind regards,

Ceren Kabukcu, PhD

Academic Editor

PLOS ONE
---

## [Editor Report · Acceptance letter]

9 Mar 2022

PONE-D-21-23001R1 

Does the Locally-Adaptive Model of Archaeological Potential (LAMAP) work for hunter-gatherer sites? A test using data from the Tanana Valley, Alaska 

Dear Dr. Driver:

I'm pleased to inform you that your manuscript has been deemed suitable for publication in PLOS ONE. Congratulations! Your manuscript is now with our production department. 

Kind regards, 

on behalf of

Dr. Ceren Kabukcu 

Academic Editor

PLOS ONE